# Resveratrol/Selenium Nanocomposite with Antioxidative and Antibacterial Properties

**DOI:** 10.3390/nano14040368

**Published:** 2024-02-16

**Authors:** Nina Tomić, Magdalena M. Stevanović, Nenad Filipović, Tea Ganić, Biljana Nikolić, Ina Gajić, Dragana Mitić Ćulafić

**Affiliations:** 1Group for Biomedical Engineering and Nanobiotechnology, Institute of Technical Sciences of SASA, Kneza Mihaila 35/IV, 11000 Belgrade, Serbia; nina.tomic@itn.sanu.ac.rs (N.T.); nenad.filipovic@itn.sanu.ac.rs (N.F.); 2Faculty of Biology, University of Belgrade, Studentski trg 16, 11000 Belgrade, Serbia; tea.ganic@bio.bg.ac.rs (T.G.); biljanan@bio.bg.ac.rs (B.N.); 3Faculty of Medicine, Institute of Microbiology and Immunology, University of Belgrade, 11000 Belgrade, Serbia; ina.gajic@med.bg.ac.rs

**Keywords:** resveratrol, selenium nanoparticles, antioxidative, antibiofilm, biocompatibility

## Abstract

In this work, we synthesized a new composite material comprised of previously formulated resveratrol nanobelt-like particles (ResNPs) and selenium nanoparticles (SeNPs), namely ResSeNPs. Characterization was provided by FESEM and optical microscopy, as well as by UV-Vis and FTIR spectroscopy, the last showing hydrogen bonds between ResNPs and SeNPs. DPPH, TBA, and FRAP assays showed excellent antioxidative abilities with ResNPs and SeNPs contributing mainly to lipid peroxidation inhibition and reducing/scavenging activity, respectively. The antibacterial effect against common medicinal implant colonizers pointed to notably higher activity against *Staphylococcus* isolates (minimal inhibitory concentrations 0.75–1.5%) compared to tested gram-negative species (*Escherichia coli* and *Pseudomonas aeruginosa*). Antibiofilm activity against *S. aureus*, *S. epidermidis*, and *P. aeruginosa* determined in a crystal violet assay was promising (up to 69%), but monitoring of selected biofilm-related gene expression (*pel*A and *alg*D) indicated the necessity of the involvement of a larger number of genes in the analysis in order to further establish the underlying mechanism. Although biocompatibility screening showed some cytotoxicity and genotoxicity in MTT and alkaline comet assays, respectively, it is important to note that active antioxidative and antibacterial/antibiofilm concentrations were non-cytotoxic and non-genotoxic in normal MRC-5 cells. These results encourage further composite improvements and investigation in order to adapt it for specific biomedical purposes.

## 1. Introduction

Since the emergence of biomaterials in tissue engineering at the end of the 20th century, there have been enormous advances in this field of biomedicine [1]. Nevertheless, some issues are still unresolved. Inflammation, oxidative stress, bacterial colonization, and biofilm formation are the most relevant problems that demand upgrading in biomaterials science [2,3]. Due to emergent antimicrobial resistance, there is a growing scientific interest in implementing alternative, nature-derived agents as multifunctional additives in tissue engineering [2]. Natural mineral or plant-derived compounds offer prospective potential for finding novel biocompatible agents that possess therapeutic potential. Simultaneously, they address the issues of bacterial infections, oxidative stress, tissue healing, etc. [2]. Furthermore, exponential progress regarding the use of such agents has been allowed with the development of nanotechnology. Substance properties can be modulated by the synthesis of nanoforms or by using various carriers to deliver them to the target site in the body. In addition to more widely used metal and polymeric nanoparticles, attention in the field of nanotechnology in recent years has been given to polyphenols and selenium compounds [4,5].

Resveratrol is a polyphenol from the stilbenoid family of phytoalexins, with a wide range of biological activities, among them antibacterial, antifungal, cardioprotective, neuroprotective, and anticancer [6,7,8]. Its low bioavailability has been an obstacle to practical use, leading to the development of nanoformulations to improve therapeutic potential [4]. Resveratrol polycaprolactone-based nanoformulations have improved the bioavailability of resveratrol and showed promise for use in ocular medicine [9], and several reported nanoformulations enhanced resveratrol’s potential as an anticancer drug [10]. In our previous research, we created carrier-free ResNPs in a shape that was especially convenient for further material design and provided preliminary information concerning antioxidant and antibacterial properties [11].

Selenium, a nonmetal essential micronutrient, exists in selenoproteins, such as the antioxidative enzyme glutathione peroxidase [12]. Its low amount is crucial for the maintenance of redox balance in the organism, but high doses induce toxic effects [13]. Different types of selenium nanoparticles have been shown to have better biocompatibility and improved therapeutic effects compared to commonly used selenium compounds [14]. Moreover, previously formulated SeNPs by our scientific group have shown antimicrobial activity and have been denoted as prosperous candidates for further biomaterials upgrading and the improvement of different biomedicinal devices [14].

The purpose of this study was to develop a novel nanocomposite material, ResSeNPs, based on previously formulated ResNPs and SeNPs. The study aimed to comprehensively characterize this compound and conduct evaluations of its antioxidative and antimicrobial properties, along with assessing its biocompatibility. All biological activities were tested for single components as well to clearly define the contribution of each to overall nanocomposite effectivity. The selection of tests and model strains was made with respect to previously published data to provide novelty.

To achieve this goal, green synthesis was used for the nanocomposite formulation. Physicochemical characteristics were determined by UV-Vis and FTIR spectroscopy, field-emission scanning electron microscopy (FESEM), optical microscopy, and a simple retention study, while antioxidative properties were tested by DPPH, TBA, and FRAP assays. Further on, the antimicrobial potential was investigated by microdilution and biofilm formation assays. Both were performed on carefully selected bacterial strains known to commonly colonize medical devices. These strains were not used in our previous research [14]. In addition, *Pseudomonas aeruginosa* strains were selected to monitor the expression of selected biofilm-related genes by the RT-PCR method to provide an in-depth analysis of the underlying mechanism. Finally, MTT and alkaline comet assays on different cell lines were used to determine biocompatibility in terms of cytotoxicity and genotoxicity, respectively.

## 2. Materials and Methods

### 2.1. Materials

Reagents were of analytical grade: trans-resveratrol (≥98%) was from ChromaDex (Irvine, CA, USA). Ethanol (96%) and methanol (99.5%) were purchased from Zorka Pharma, Sabac, Serbia. Mueller Hinton broth was obtained from Torlak, Belgrade, Serbia. Crystal violet and streptomycin were from Thermo Fisher Scientific, (Waltham, MA, USA). Sodium selenite, bovine serum albumin (BSA), Dulbecco’s modified eagle medium, phosphate-buffered saline (PBS), fetal bovine serum (FBS), penicillin-streptomycin mixtures, trypsin from porcine pancreas, dimethyl sulfoxide (DMSO), trichloroacetic acid, Triton^®^ X-100, normal melting point agarose (NMP), low melting point agarose (LMP), EDTA, Tris, NaOH, potassium dihydrogen phosphate, disodium phosphate, sodium phosphate monobasic, iron (II) sulfate, iron (III) chloride, potassium ferricyanide, thiobarbituric acid, perchloric acid, 3-(4,5-dimethylthiazol-2-yl)-2,5-diphenyltetrazolium bromide (MTT), acridine orange, resazurin sodium salt, and 2,2–diphenyl-1-picrylhydrazylhydrate (DPPH) were purchased from Sigma-Aldrich (St. Louis, MO, USA). Commercial preparation of liposomes ‘‘PRO-LIPO S” pH 5–7 was from Lucas–Meyer, Hamburg, Germany. Ascorbic acid was obtained from BDH Prolabo, (VWR Chemicals, Radnor, PA, USA). TRIzol reagent was purchased from Ambion by Life technologies (Thermo Fisher Scientific, Waltham, MA, USA). Distilled water and DNaseRNase Free was from Gibco by Life technologies, (Thermo Fisher Scientific, Waltham, MA, USA. Chloroform was from Fisher Chemicals, (Thermo Fisher Scientific, Waltham, MA, USA) and isopropanol was from Carlo Erba Reagents, (Emmendingen, Germany).

### 2.2. Synthesis of ResSeNPs Composite

ResNPs were synthesized by the physicochemical solvent–nonsolvent method, as described in Tomić et al. (2023) [11]. Briefly, the trans-resveratrol powder was dissolved in 96% ethanol (50 mg/5 mL) and, during 5 min of homogenization at 17,500 rpm, this solution was drop-wise added into 25 mL of precooled water. The resulting particle suspensions were stored at 5 °C, under protection from light.

For the synthesis of SeNPs, we used the method described in Filipovic et al. Sodium selenite was dissolved at a concentration of 0.02 M in distilled water. Solution of ascorbic acid of 0.125 M and BSA of 0.87% *w*/*w* were mixed and stirred at low speed for 2 min using a magnetic stirrer. Then, while stirring at 1500 rpm, a sodium selenite solution was dropwise added into the mixture, leading to the appearance of a red-orange color, indicating the formation of selenium nanoparticles. The sodium selenite and BSA mass ratio was 1:1. Concentration of the SeNPs was in further experiments expressed as the concentration of selenium, determined by inductively coupled plasma mass spectrometry analysis, although the other components (excess ascorbic acid and BSA) also remained. Reaction mixtures were protected from light, preventing crystallization and photo-oxidation of SeNPs, before storing at 5 °C [14].

ResSeNPs were obtained by combining the suspension of ResNPs (adjusted to 1600 µg/mL) and colloidal solution of SeNPs (adjusted to 600 µg/mL of selenium). Five mL of each was combined and firstly homogenized at 17,500 rpm for 1 min, and then at 9500 rpm for 4 min (Figure 1). During high-speed homogenization, an intensive, uniform orange color was developed in under 1 min. The obtained ResSeNPs composite suspension was stable and homogenous, with a starting concentration of 800 µg/mL of ResNPs and 300 µg/mL of SeNPs. The sample was subsequently incubated at 37 °C for 6 h. pH measurements were conducted immediately after synthesis and after 6 months (storage temperature being 5 °C).

### 2.3. UV-Vis Spectroscopy

UV-Vis absorbance of freshly prepared ResSeNPs samples and samples stored for 6 months was measured on a GBC Cintra UV-Vis spectrophotometer at room temperature, in the wavelength range 200–700 nm, with a standard quartz cuvette cell path (10 mm).

### 2.4. Fourier Transform Infrared Spectroscopy (FTIR)

FTIR spectroscopy was used for qualitative analysis of the samples. FTIR spectrum of ResSeNPs, along with ResNPs and SeNPs for comparison, was recorded at Nicolet iS10 (Thermo Fisher Scientific, Waltham, MA, USA) spectrometer, in attenuated total reflectance (ATR) mode. Measurements were performed in a spectral range of 400–4000 cm^−1^ with a resolution of 4 cm^−1^, with 32 scans.

### 2.5. Optical Microscopy

The surface and morphology of ResSeNPs samples were observed by OPTICA B-500MET light microscope (Optica SRL, Ponteranica, Italy) with OPTIKAM PRO 8LT–4083.18 camera equipped with a scientific-grade CCD sensor, under 200, 500, and 1000× magnification, using Optica Vision Pro software (2014, 4.4, Optica SRL, Ponteranica, Italy).

### 2.6. Field-Emission Scanning Electron Microscope (FESEM)

The scanning electron microscope Tescan Mira 3 XMU FE-SEM (Brno, Czech Republic) operated at 20 keV was used to get a closer insight into the morphology of the ResSeNPs composite. The samples were applied at the metallic stub, air-dried for 24 h, and sputter-coated with a thin layer of gold before imaging.

### 2.7. SeNPs Retention Study

In order to determine how much of the SeNPs were retained on the ResNPs, we used filtration to separate free SeNPs. Samples were then centrifuged and washed to remove excess ascorbic acid and BSA, resuspended in distilled water and their absorbance measured in the wavelength range 200–400 nm. As a control, we performed the same procedure using the starting SeNPs suspension, diluted two-fold for proper comparison of concentrations. Absorbance values of established concentrations of SeNPs were then used for the determination of the concentration of released SeNPs. To estimate the dynamics of the SeNPs release from ResSeNPs, we incubated ResSeNPs samples for 12 h at 37 °C to mimic the conditions of biological tests. The amount of free SeNPs was then compared to the free SeNPs amount measured immediately after the synthesis.

### 2.8. Determination of Antioxidative Activity

#### 2.8.1. 2,2–Diphenyl-1-picrylhydrazylhydrate (DPPH) Reduction Assay

DPPH is one of the most used and convenient methods to assess the antioxidative activity of test substances [15]. It is a colorimetric assay, in which DPPH (2,2–diphenyl-1-picrylhydrazylhydrate), a stable nitrogen radical, accepts hydrogen atoms from the antioxidative compounds. This leads to the color change from the original purple to yellow, recorded as the decrease of DPPH absorbance peak at 517 nm. ResNPs, SeNPs, and ResSeNPs were dispersed in methanol. Ascorbic acid was used as a positive control, while the experimental mixture without a test sample was used as a negative control. DPPH stock of 200 µM was prepared in methanol with the use of ultrasound and protected from light. To 800 µL of each sample, 200 µL of DPPH stock was added, followed by 30 min of incubation and measurement of absorbance at 517 nm GBC Cintra UV–Vis spectrophotometer. The final concentration ranges of ResNPs, SeNPs, and ResSeNPs were 1–10 µg/mL, 0.1–5 µg/mL, and 0.39–1.5%, respectively.

DPPH scavenging activity was calculated using the formula:Scavenging activity (%) = 100 * (A_C_ − A_t_)/A_C_

In the formula, A_C_ is the A_517_ value of the control sample, representing the absorbance of non-reduced DPPH after 30 min of incubation, and A _t_ was the absorbance of the test sample.

#### 2.8.2. Effect on Lipid Peroxidation (TBA Assay)

A TBA assay was used to determine the influence of ResNPs (1.125–5 µg/mL), SeNPs (0.5–2 µg/mL), and ResSeNPs (0.185 and 0.39%) on lipid peroxidation. MDA, a product of lipid peroxidation, reacts with thiobarbituric acid, which leads to the formation of a colored adduct, whose absorbance is measured at 532 nm. The assay was performed by the protocol given by Mitić-Ćulafić et al. (2009) [16]. Commercial liposomes were used as a model of biological membranes. Lipid peroxidation was induced by adding 10 µL of 0.01 M FeSO_4_ and 20 µL of 0.01 M ascorbic acid to 60 µL of 10% *w*/*v* liposomes, with or without 10 µL of test substances. One series of samples was prepared without liposomes (correction) and one series without a test substance (control). Then, the 0.067 M buffer was added and samples were incubated at 37 °C. After 1 h, 2 mL of TBA reagent and 0.2 mL of 0.1 M EDTA were added, followed by incubation at 100 °C for 15 min. Samples were then cooled and centrifuged at 4000 rpm for 10 min. The absorbance of supernatant was measured at 532 nm (GBC Cintra UV–Vis spectrophotometer), and obtained values used to calculate the percent of inhibition of lipid peroxidation (I):I (%) = (1 − (At − Acr)/Ac) * 100

In the formula, At represents the absorbance of the test sample, Acr represents the absorbance of the correction, and Ac is the absorbance of the control.

#### 2.8.3. Ferric Cyanide (Fe^3+^) Reducing Antioxidant Power Assay (FRAP Assay)

FRAP assay measures the potential of substances to reduce iron ions. Transfer of electrons from antioxidant to Fe^3+^ oxidant leads to reduction to Fe^2+^, a reaction visible by color change from yellow to green and blue. Briefly, as described by Cvetković et al. (2021) [17], samples were prepared in methanol before adding the phosphate buffer and 1% potassium ferricyanide, with final concentration ranges being 1–10 µg/mL of ResNPs, 0.1–5 µg/mL of SeNPs and 0.185–1.5% of ResSeNPs. After 20 min of incubation at 50 °C, trichloroacetic acid was added, and the samples were centrifuged. The researchers mixed 2.5 mL of supernatant with 2.5 mL of water and 0.5 mL of 0.1% FeCl_3_, and absorbance was measured at 700 nm [15] (GBC Cintra UV–Vis spectrophotometer). Ascorbic acid was used as a positive control, while the experimental mixture without a test sample was used as a negative control.

### 2.9. Determination of Antibacterial Activity

To investigate the antimicrobial activity, five ATCC and four bacterial isolates obtained from hospitalized patients were provided by the Institute for Microbiology and Immunology, University of Belgrade—Faculty of Medicine, Belgrade. All gram-positive and gram-negative bacteria are listed in Table 1. Bacterial cultures were maintained on agar plates at 5 °C for 2 weeks or in frozen stocks in liquid nitrogen for longer periods.

#### 2.9.1. Microdilution Assay

All strains were inoculated in Mueller Hinton broth (MHB) and incubated overnight at 37 °C. Using OD_600_ measurements, the number of bacteria was adjusted to 10^8^ cells/mL and then diluted to 10^6^ cells/mL in 0.01M MgSO_4_. Twenty μL of suspension, containing 2 × 10^4^ of bacterial cells was inoculated to each well of the microtiter plate and exposed for 24 h to serial dilutions of ResNPs (6.25–800 μg/mL), SeNPs (4.67–300 μg/mL, and additional 400 μg/mL) and ResSeNPs suspensions (0.37–50%). Streptomycin was used as an antibiotic control. Color controls of ResNPs, SeNPs, and composite were also prepared using serial dilutions of test substances without bacteria.

After incubation, resazurin was added to each well, and plates were incubated for 2 h until the control wells showed a full reduction of resazurin to pink product resorufin. Samples from the wells with no observable change of color were inoculated on Mueller Hinton agar plates and incubated overnight at 37 °C to determine minimal bactericidal concentration.

#### 2.9.2. Antibiofilm Activity

For the monitoring of the antibiofilm properties of ResNPs, SeNPs, and composite, the preparation of bacteria was conducted as in minimal inhibitory concentration assays, but 10× more bacteria were seeded to every well containing various concentrations of samples. Glucose was added up to 2% to promote biofilm formation. Bacteria were then incubated overnight, followed by the removal of the medium and washed 3 times with 1xPBS to remove all non-adherent bacteria. The researchers added 0.1% crystal violet solution to the wells and incubated for 20 min. Then, the plates were rinsed with fresh water to remove excess stains and thoroughly dried. The stain left bonded to biofilms was dissolved by using absolute ethanol, and absorbance was recorded at 570 nm using a Multiskan plate reader (Thermo Fisher Scientific, Waltham, MA, USA).

#### 2.9.3. Reverse Transcription-Polymerase Chain Reaction (RT-PCR) Analysis

Gene expression analysis was conducted by following the expression of *alg*D and *pel*A genes in *P. aeruginosa* reference strain and clinical isolate. The presence of the selected genes was confirmed by PCR analysis. Bacteria were cultivated overnight, and the set up was conducted as in the antibiofilm assay, only in 12 wells plates, for cultivating larger quantities of biofilm. Treatment consisted of 12 µg/mL of ResNPs, 4.7 µg/mL of SeNPs, and 1.5% of ResSeNPs. One group of wells was left untreated and served as a control. Plates were then incubated overnight at 37 °C.

##### RNA Isolation and cDNK Library Preparation

Firstly, the medium was removed and wells were washed with warm 1xPBS. Then, a Trizol reagent was used to dissolve other cell components, excluding RNA. Chloroform was then added, before high-speed centrifugation, to separate the aqueous and organic phases. RNA was then precipitated and cleaned by using ice-cold isopropanol and ethanol and centrifugation. Samples were then thoroughly dried, resuspended in 50 µL of PCR water, and then incubated for 5 min at 60 °C before storing at −80 °C. Obtained RNA concentrations were checked by measuring relations in 260/280 and 260/230 nm absorbances, by using a NanoDrop 2000c spectrophotometer (Thermo Fisher Scientific, Waltham, MA, USA).

Transcription of RNK into the cDNK molecules was performed in 20 μL volume, following the manufacturer’s protocol from kit Revert Aid First Strand cDNA Synthesis Kit, (Thermo Fisher Scientific, Waltham, MA, USA).

Expression of the previously mentioned genes under the influence of the treatments was measured using quantitative RT-PCR on Step One Plus, Real Time PCR System (Applied Biosystem by Thermo Fisher Scientific, Waltham, MA, USA). Each PCR reaction was prepared in a total volume of 12 μL, containing cDNA (30 ng), selected primers (500 nM) (Table 2), and PowerUp SYBR Green PCR Master Mix (Applied Biosystems by Thermo Fisher Scientific, Waltham, MA, USA). As an endogenous control was used, 16S rRNA and all the samples were normalized compared to it. Reaction conditions were as follows: 50 °C for 2 min, 95 °C for 10 min, followed by 40 cycles on 95 °C for 15 s and after 60 °C for 1 min. According to the previously described [18], the cycle threshold (Ct) method was used to calculate relative gene expression.

### 2.10. Determination of Biocompatibility

#### 2.10.1. Cell Culture

Cell lines used for cytotoxicity and genotoxicity study were the human normal lung fibroblast MRC-5, human lung adenocarcinoma alveolar basal cell A549, and human hepatocellular carcinoma HepG2 cell lines. All three cell lines were adherent. MRC-5 cells are recognized as a standard model in testing the cytotoxicity of medical devices, as recommended by ISO 10993-5:2009(E) [22]. A549 cells, being from the same organ, are often used for comparison with MRC-5 as a model for differential substance effects on normal and cancerous cells. HepG2, a liver-derived cell, offers reliable toxicology testing because of its active enzymes, which simulate processes in the liver. MRC-5, A549, and HepG2 cells were from the ATTC collection, ECACC 84101801, ATCC CCL-185, and ATCC HB-8065, respectively.

Cells from liquid nitrogen storage were defrosted and immediately transferred to cell media. The remaining DMSO was removed by centrifugation followed by the change of media. Cells were cultivated until confluency in the Dulbecco’s modified Eagle media supplemented with 5% fetal bovine serum (or 10% in case of HepG2) and 100 U/mL penicillin/streptomycin mix, in a humidified incubator with 5% CO_2_, at 37 °C. Trypsinization with 0.1% trypsin-EDTA was used for passaging the cells before reaching the number of cells sufficient for the experiments.

#### 2.10.2. Determination of Cell Viability by Using MTT Assay

Cytotoxicity of ResNPs, SeNPs, and ResSeNPs was determined by using MTT dye, 3-[4,5-dimethylthiazol-2-yl]-2,5-diphenyltetrazolium bromide. Firstly, 2 × 10^4^ cells were seeded in 200 µL of DMEM in each well of 96 well plates. Plates were incubated at 37 °C for 24 h, followed by the removal of the medium, and the addition of fresh medium containing adjusted concentrations of test substances. ResNPs were tested in the range 3.125 to 400 µg/mL, SeNPs 4.68-300 and 400 µg/mL, and ResSeNPs suspension from 0.37% to 50% of suspension. Cells were exposed to the samples for the next 24 h, after which the medium was discarded, followed by careful washing of the cells using 1xPBS. Then, 200 µL of new medium containing 0.5 mg/mL of MTT dye was added to each well, and plates were then incubated for 3 h, until the formation of formazan crystals derived from MTT in a process occurring in proportion to the number of viable cells. Medium containing MTT was then removed and formazan crystals dissolved in DMSO followed by measurement of absorbance at 570 nm (Multiskan plate reader).

#### 2.10.3. Assessment of Genotoxic Effect on Human Cells by Alkaline Comet Assay

For the genotoxicity study, 3 × 10^5^ cells were seeded in each well in 12 well plates and incubated until the formation of a monolayer. Then, the medium was removed and cells were washed with 1xPBS. Three non-cytotoxic concentrations of every sample were chosen for the treatment, for a duration of 24 h. On the day of the experiment, treatment with 0.1% hydrogen peroxide for 15 min was used as a positive control before the trypsinization of the cells, followed by other steps in the sample preparation and alkaline comet assay procedure, performed as described in detail by Nikolić et al. (2011) [23]. Staining of the samples was conducted using 2 µg/mL of acridine orange, and individual nuclei were observed by microscope by using Comet Assay IV Software (Perceptive Instruments, Bury St Edmunds, UK). Fifty nuclei per gel (100 cells/slide, in every replicate) were analyzed. Results were expressed via Tail Intensity as a parameter of DNA damage, with Tail Intensity representing the percent of DNA in the tail of the comet.

### 2.11. Statistical Analysis

All of the biological experiments were conducted in triplicate to ensure reliability, followed by data processing and analysis using OriginLab Software (OriginLab Corporation), Microsoft Excel (Microsoft, Redmond, WA, USA), or GraphPad Prism 5 (GraphPad Software, Dotmatics, 2023, 9.0, Boston, MA, USA). Results were expressed as the average of the obtained results, +/− standard deviation. The Student’s *t*-test and one-way ANOVA followed by the Tukey post hoc test were used to analyze the results of antioxidative assays, CV, and MTT tests. Results obtained from quantitative RT-PCR were analyzed using a one-way ANOVA with Dunnet’s post hoc test. The level of statistical significance was * *p* < 0.05. Analysis of the statistical significance of comet assay results was conducted by the Mann-Whitney U test, and significance levels were represented as follows: * *p* < 0.05; ** *p* < 0.01.

## 3. Results

ResSeNPs were made by combining previously characterized ResNPs and SeNPs in the ratio of 1:1, containing them in concentrations of 800 μg/mL and 300 μg/mL, respectively. The obtained composite was assessed in terms of its physicochemical characteristics, while selected biological activities were determined for both the composite and its components. Samples were used in the activity experiments for up to 1 month after synthesis.

Macroscopically, ResSeNPs were homogenous suspensions of bright orange color, becoming darker during storage (Figure 2, insert). Measured pH values are represented in Figure 2 (insert), showing that during the 6 months of storage, there has been a decrease in the pH values of all samples.

### 3.1. UV-Vis Analysis

UV-Vis spectra are displayed in Figure 2. The measurement of ResNPs absorption, immediately after synthesis and after 6 months, showed a broad absorption peak of identical intensity at 310 nm, while in the case of SeNPs, the peak was positioned at around 265 nm and was less pronounced in the stored sample. The measurement of fresh ResSeNPs absorption showed a wide peak at 310 nm as well as a peak at 265 nm, but in the stored sample, the 265 nm peak exhibited a significant decrease.

### 3.2. FTIR Analysis

FTIR spectroscopy was performed to evaluate the possible interaction between ResNPs and SeNPs in ResSeNPs. FTIR spectra are represented in Figure 3. As mentioned earlier, SeNPs were stabilized with BSA and thus possess complex chemical structures, rich with chemical groups capable of interacting with hydroxyl groups in resveratrol. By comparing the spectra of SeNPs, ResNPs, and ResSeNPs, there is a significant difference in the intensity and the shape of the broad peak from the higher wavenumber region, around 3250 cm^−1^, which corresponds to stretching vibrations of OH and NH functional groups. This peak was more intense and wider in the composite sample, indicating possible hydrogen bonding between resveratrol and BSA from the surface of SeNPs. In addition to this, the composite exhibits a low-intensity peak originating from carbonyl stretching vibration around 1780 cm^−1^ recorded in the SeNP spectrum as well. Resveratrol-characteristic bands in lower wavenumber region, such as multiple peaks in ranges of 1610–1440 cm^−1^ (benzene rings), 1150–1010 cm^−1^ (vibrations of C-O bonds), or 980–830 cm^−1^ (out-of-plane bending of vibrations of C-H bonds) could be observed in the composite sample as well, with a small intensity difference. This difference could be correlated with the overlapping of some peaks in SeNPs and ResNPs.

The spectra of SeNPs in the lower wavenumber region featured bands from the SeNP components: C–C and C–O (around 1050 cm^−1^), bending vibrations of C–H (1350 cm^−1^), and a band at 1580 cm^−1^ specific for stabilized SeNPs. All these bands are present in the composite’s spectra but in a slightly shifted position.

### 3.3. Optical Microscopy

The direct visualization and imaging of materials under ambient conditions hold immense significance in both their characterization and practical application. This is vital as the morphology plays a crucial role in determining how the material interacts with cells and tissues, thereby influencing its potential efficacy for biomedical applications. In Figure 4A, a representative image displays uniform, interlaced nanobelt-like ResNPs. However, in Figure 4B, optical microscope images reveal agglomerates of SeNPs associated with elongated particles of ResNPs. Notably, images C and D demonstrate that their morphology remained unchanged even after exposure to an elevated temperature of 37 °C. This persistence of morphology suggests the stability of the composite material.

The size of the ResNPs was 17 µm in length and 750 nm in width, as was also previously reported [11]. SeNPs were only visible as the agglomerates, because their individual size was less than 100 nm, as determined in the previous paper [14].

### 3.4. Field-Emission Scanning Electron Microscope (FE-SEM)

FESEM imaging (Figure 5) gave closer insight into the composite material morphology. Resveratrol nanobelt-like structures were visible on FESEM micrographs. From the FESEM images (Figure 5A), it can be seen that ResNPs have a belt-like shape, elongated morphology, and width below one micron, i.e., nanoscale dimension. The ResNPs exhibited a large ratio of width to length, resembling nanobelts. The mean the width of ResNPs, as measured by ImageJ software from FESEM micrographs, was around 750 nm. The particles displayed uniformity, with a tendency to group into larger clusters or bundles. These structures were unaltered regarding their morphology and dimensions in the composite sample.

On the other hand, SeNPs from the sample of the original SeNP suspension could be seen as the agglomerates, which was probably caused by the drying process (Figure 5B). Nevertheless, single particles were still observable. It can be seen that the SeNPs are spherical, uniform, and have a narrow particle size distribution (Figure 5B), with a particle diameter below 100 nm (Figure 5C). This is in accordance with the data from the paper presenting SeNPs made by this method, where it was shown that they were amorphous, of spherical shape, and 70–100 nm in size [14]. 

In the composite samples, under the lower magnification, there was a visible increase of roughness of ResNPs in the composite sample, caused by the coating by SeNPs. Under the higher magnification, single SeNPs were visible scattered on the surface of ResNPs, mostly uniformly (Figure 5C). 

### 3.5. 2.7 SeNPs Retention Study

Results regarding the amount of free SeNPs from the ResSeNPs are presented in Table 3. Results are expressed in percentages of SeNPs, compared to the concentration of the SeNPs added to the composite mixture, ± variation between the replicates. It can be seen that immediately after the homogenization, approximately one-third of the SeNPs added to the composite were not attached. The difference in the proportion of attached and free SeNPs was not statistically significant, during the 12 and 24 h of incubation.

### 3.6. Antioxidative Activity

Results obtained in the DPPH assay pointed out that both composite and single components had high antioxidative activity, but the efficacy of ResNPs was somewhat lower (Figure 6).

Results obtained in the TBA assay showed a significant inhibitory effect of the ResSeNPs composite on the process of lipid peroxidation (Figure 7). Interestingly, the overall activity (up to 80%) was remarkably higher than the activity of ResNPs (dose-dependent, up to 40%) and SeNPs (up to 30%, but with inverse dose dependency).

Concerning FRAP assay, the synergistic effect of ResNPs and SeNPs was also indicated (Figure 8). While the activity of ResNPs and SeNPs reached 0.39 and 3.18 values of (A_700_), respectively, ResSeNPs induced multiply higher activity (A_700_ up to 6.63).

### 3.7. Antibacterial Activity

#### 3.7.1. Microdilution Assay

Results obtained in the microdilution assay performed on selected bacterial strains (Table 4) showed the absence of activity or weak activity against gram-negative strains, while activity on gram-positive was notably higher. Concerning gram-positive bacteria, a comparison of a single component’s efficacy indicated the higher effect of SeNPs, being even enhanced in the case of ResSeNPs composite effect on some of the tested strains (*S. aureus* GP41, both *S. epidermidis*). Moreover, among gram-negative bacteria, a composite mixture even sensibilizes referent *E. coli* strain to SeNPs.

#### 3.7.2. Antibiofilm Activity

Antibiofilm activity was tested on the selected species relevant for infections established due to medicinal implant colonization. Results obtained by the CV assay (Figure 9), showed that ResNPs had an inhibitory effect on biofilm formation of *S. aureus* ATCC29213, both *P. aeruginosa* strains. In addition, some activity was noted on the *S. epidermidis* isolate. SeNPs showed notable activity against all strains except the *S. epidermidis* referent strain. Compared to single components, ResSeNPs had better inhibitory effects in the majority of tested bacteria. Results obtained on *P. aeruginosa*, especially on the clinical isolate, were most noteworthy. ResSeNPs, as well as its constituents, possessed antibiofilm activity up to 49.95%, 69.81% and 55.15% of inhibition of biofilm biomass at 100 µg/mL of ResNPs, 18.75 µg/mL of SeNPs, and 6.25% of ResSeNPs, respectively.

#### 3.7.3. RT-PCR Analysis

To deepen the analysis of the antibiofilm activity against the most susceptible *Pseudomonas* strains, the expression of selected genes (*alg*D and *pel*A) involved in its biofilm formation was monitored. Results showed an opposite tendency of activity in referent strain and isolate, i.e., lowering and increasing, respectively (Figure 10). Concerning expression inhibition in the ATCC strain, the most pronounced activity was observed for ResSeNPs. However, the absence of statistical significance pointed out that there was no clear influence on selected gene expression.

### 3.8. Biocompatibility of ResSeNPs

#### 3.8.1. MTT

An MTT assay was performed on the MRC-5, A549, and HepG2 cells and the results are shown in Figure 11, showing cell survival dynamics and IC50 values. A549 and MRC-5 showed similar sensitivity to ResNPs, while ResSeNPs and especially SeNPs were notably less toxic for MRC-5 cells. Concerning single constituents’ treatment, it is important to note that normal MRC-5 fibroblasts were the most sensitive to ResNPs, but least sensitive to SeNPs. The same trend of higher toxicity for cancer cells and less toxicity for MRC-5 cells, observed for SeNPs, is preserved in ResSeNPs composite treatments. Comparison of ResSeNPs activity on one hand, and each single treatment activity on the other, showed that overall composite cytotoxicity is lower than the simple sum of cytotoxicity of each component applied alone, indicating some antagonism in the normal cell line. In other words, in the case of low tested concentrations, minor toxic SeNPs attenuated the high toxicity of ResNPs in composite material. In cancer cells A549 and HepG2, this effect was non-significant or even inverse, respectively.

#### 3.8.2. Comet Assay

Alkaline comet assay results on the MRC-5, A549, and HepG2 cell lines are shown in Figure 12. ResNPs showed low genotoxicity in A549 and HepG2 cell lines, but in MRC-5 it led to a lower level of DNA damage compared to the control. SeNPs induced low-level genotoxicity in all cell lines. Most importantly, ResSeNPs did not show genotoxicity in MRC-5 as well as in A549, while in HepG2, the highest concentrations of ResSeNPs induced slight DNA damage.

## 4. Discussion

UV-Vis spectroscopy of the newly prepared composite confirmed the presence of peaks characteristic of individual components. A peak at 310 nm confirmed that resveratrol remained in the trans-isoform in both ResNPs and ResSeNPs during the 6 months of storage, indicating the chemical stability of this component, solely or in the composite. On the other hand, the decrease of the 265 nm peak during the long storage, indicated decomposition of some of the components of SeNPs or a chemical modification in the ResSeNPs composite.

From our previous paper, SeNPs zeta potential was +27 ± 3 mV, indicating good stability in the suspension [14]. However, this parameter could not be experimentally determined for ResNPs and ResSeNPs due to the length of ResNPs.

The decrease of pH in the stored samples could be related to the observed absorbance phenomenon. As all of the following experiments were conducted by using samples for no longer than a few weeks, possible chemical interactions, decomposition, and stability in time will be further assessed in a separate study.

FTIR spectroscopy was used to assess the binding of components. The shifting in characteristic bands in FTIR spectra indicated the formation of the new hydrogen bonding in ResSeNPs. Xiao et al. (2008) [24] showed that trans-resveratrol spontaneously binds with BSA based on hydrogen bonds and van der Waals interactions. This was further supported by Bourassa et al. (2010) [25], who in addition determined partial protein unfolding during this binding. It is, therefore, our conclusion that the interaction between ResNPs and SeNPs occurs via the interaction of resveratrol and BSA, the stabilizer component of SeNPs. The images obtained by optical microscopy and SEM further support this observation. SeNPs could not be seen on the optical microscope due to their size, but the structures attached to the ResNPs most likely present SeNPs agglomerates formed because of the interaction of the ResNPs surface and BSA component of SeNPs. Furthermore, SEM micrography showed a coating of ResNPs in the composite sample. The structure of this coating changed to more differentiated during prolonged exposure to physiological temperature, possibly due to continuous BSA unfolding initially caused by binding to resveratrol [25] or due to the other changes and interactions between SeNPs. Furthermore, FESEM micrography showed the attachment of SeNPs to ResNPs in the composite sample. Moreover, single SeNPs can be seen attached to the surface of ResNPs on the micrographs obtained under higher magnifications.

The retention study showed that approximately 65% of the SeNPs became associated with the surface of ResNPs after homogenization. This parameter did not significantly change during the incubation for 24h. However, due to the fact that the SeNPs were on the surface of ResNPs, both attached and released SeNPs could contribute to the observed biological effects.

ResSeNPs, as well as their components, were first tested regarding their antioxidative potential. It is known that the most notable exogenous antioxidants are ascorbic acid, polyphenolic compounds, and minerals: selenium and zinc [26]. This indicated that resveratrol (polyphenol) and selenium combinations could be used in our design of a multifunctional composite material, or as additional components of biomaterials, to provide them with antioxidative potential. It is also known that a combination of antioxidative compounds results in even better long-term effects than the use of a single antioxidant [15]. Notable antioxidative activity of ResSeNPs was shown by DPPH, TBA, and FRAP assays. ResSeNPs components possess high antioxidative activity based on both constitutive components; however, obtained results pointed out that the SeNPs contributed dominantly to the overall antioxidative activity. Moreover, SeNPs suspension contains ascorbate which also could contribute to overall antioxidant potential. The observed antioxidative effect was achieved by multiple antioxidative mechanisms. It is already known from the literature that, owing to the hydroxyl group on the benzene ring and a system of a conjugated double bond, resveratrol has a high capability of scavenging free radicals and is thus a potent antioxidant [27]. It also has Fe^2+^ chelating ability, reducing power and H_2_O_2_ scavenging activity [28]. Although ResNPs were less efficient compared to SeNPs in terms of direct antioxidative effects, they showed concentration dependency, significant inhibition of lipid peroxidation at low concentrations, and synergy with SeNPs in TBA and FRAP. On the other hand, SeNPs suspension had high activities in both DPPH and FRAP and no effect on lipid peroxidation. The observed synergy in the tests could be originating from the connected chemical reactions of multiple components in the ResSeNPs composite, but these chemical interactions should be assessed by further tests.

All of this means that the potential of ResSeNPs for further use is promising, as oxidative stress is an important factor in tissue engineering. It influences the mineralization and degradation of bone scaffolds and inflammatory response. It has been shown that the addition of antioxidative compounds, such as polyphenols, to the bone scaffolds greatly improves redox balance and, consequently, the health of surrounding cells [2]. Resveratrol and selenium can be found in various multicomponent pharmaceutical products, but the combined antioxidative effects and effects on gene expression of their bulk forms have only recently been studied in more depth by Cosín-Tomàs et al. (2019) [29]. Furthermore, Yang et al. (2018) [30] synthesized selenium nanoparticles coated with resveratrol that have been successful in inhibiting the aggregation of amyloid β and the formation of reactive oxygen species in the brain. These findings support the idea that a combination of these agents can provide several biological benefits.

Bacterial infections caused by resistant strains are becoming more widespread, and consequently, the antibacterial effect is becoming one of the most demanded traits in biomaterials. Exploring the antibacterial potential of nanoparticles seems to be prospective since they could be featured with multiple antimicrobial mechanisms, which is challenging for antimicrobial resistance development [31]. As some of the members of the genus *Staphylococcus* present major threats to medicinal implants [32], the two most important species were focused on, *S. aureus* and *S. epidermidis*. Gram-negative *P. aeruginosa* and *E. coli* are also known to cause contamination of medicinal material and subsequent tissue infection. In our previous work, we described the effect of ResNPs with MIC values being 800 µg/mL for staphylococci and *Pseudomonas* clinical isolate, and more than 800 µg/mL for other bacteria [11]. In this work, we further investigated the antimicrobial activity of SeNPs and ResSeNPs and the formulated composite ResSeNPs. Our result pointed out that, in the case of *S. aureus* and *E. faecalis* reference strains, MIC values were detected for the compositions that contain an equal concentration of SeNPs as the MIC values of SeNPs. Based on this fact we proposed that the direct antibacterial effect of ResSeNPs on gram-positive bacteria was mostly mediated by SeNPs. This is in line with previously published data showing that selenium nanoparticles generally had a strong effect against gram-positive bacteria [14]. Moreover, when observing all *Staphylococcus* strains, it could be noticed that active concentrations of SeNPs in MIC values of composite and single treatments were multifold lowered. The only exception is the concentration of SeNPs in MIC determined for the *S. aureus* referent strain, which is of the same value. This pointed to a synergy, i.e., the potential of ResNPs to remarkably sensibilize these bacteria to SeNPs. It is of special importance that the most notable effect was observed on the members of the genus *Staphylococcus*, which are known to be responsible for most implant failures [3]. Similar results were obtained in the study of Truong et al. (2021) [33], who showed the antibacterial synergy of another polyphenol (quercetin) and acetylcholine with selenium nanoparticles in the nanocomposite. Furthermore, the suggested mechanism for their nanocomposite was based on the membrane-damaging activity of selenium nanoparticles. It is important to note that, in addition to selenium, the SeNPs suspension may also contain a residue of protein-BSA and ascorbic acid, both of which could have influenced the observed effects [34,35], but further investigation is necessary to resolve this question. In implant infections, it is usually that biofilm formation is considered as a cause of chronic inflammation [3]. In our study, the antibiofilm effect was tested by using the bacteria that were the most known biofilm producers/implant colonizers, i.e., strains of Staphylococcus species and *Pseudomonas aeruginosa*. Concerning single treatments with each component, one could note that some regularity could be observed: in the case of all isolates, SeNPs’ efficiency was higher than ResNPs’ efficiency, while ResNPs were more active against *S. aureus* and *P. aeruginosa* referent strains. Furthermore, although not active against *S. aureus* GP41, ResNPs enhanced the activity of SeNPs against this bacterial strain biofilm. Finally, if one analyzes the type of interaction between composite constituents, it could be observed that in the case of *P. aeruginosa* ATCC 27853, *S. aureus* GP41, and *S epidermidis* ATCC 12,228 (at the highest applied concentration), enhanced activity was observed. However, although antagonism was shown in the case of *P. aeruginosa* GN998, the efficiency of ResSeNPs was still notable and statistically significant. Despite observed variation between bacterial strains, the general comment could be a notable increase in the efficiency of the ResSeNPs composite compared to both single components, at least in most cases. Generally, obtained antibacterial and antibiofilm effects could be explained by the damage to the cell membranes, either by synchronic or successive action of the components.

Concerning observed antibiofilm activity and analysis of each constituent contribution, it is important to note that resveratrol itself has been shown to possess the ability to inhibit biofilm formation [36]. In addition, several types of selenium nanoparticles have also shown antibiofilm activity, and a synergistic effect with other compounds, among them antibiotics [33]. All of these indicate the notable potential of these agents to be used in combinations against biofilm development.

Based on the results of RT-PCR analysis, the expression of chosen biofilm-related genes in *Pseudomonas* strains was not significantly changed, although there was a tendency of decreased and increased expression of both monitored genes (*pel*A and *alg*D) in the reference strain and isolate, respectively. Although it is difficult to discuss such a result, one could note that disruption of biofilm (observed in crystal violet assay) by different mechanisms could lead to activation of genes in clinical isolate, aiming to produce new biofilm and to protect from the stress-production activity of tested substances. This is based on the literature data showing that subinhibitory concentrations of certain stressors (antibiotics) can lead to increased biofilm production [37,38].

The genes selected for expression monitoring are involved in the production of important *Pseudomonas* biofilm exopolysaccharides. The *pel*A is a part of the *pelA*–*pelG* operon involved in the synthesis of Pel polysaccharide, an intercellular adhesin important for the formation and sustenance of biofilm. This gene actually codes for polysaccharide deacetylase [39]. The *alg*D is a part of *algACD* operon which controls the synthesis of another important *Pseudomonas* exopolysaccharide alginate. The role of *alg*D is to code for AlgD protein, a GDP-mannose dehydrogenase that catalyzes the production of GDP-mannuronic acid, a precursor of exopolysaccharide alginate [40]. Still, obtained results, i.e., opposite effects in tested strains and absence of statistical significance, pointed out that for a clear explanation of the observed antibiofilm effect in crystal violet assay, further analysis involving more biofilm-related genes is necessary.

Determining the levels of cytotoxicity is the first step for assessing biocompatibility. MTT assay established a range of non-cytotoxic concentrations of ResSeNPs and their components. In normal cells, the levels of viability inhibitions of cells treated with ResSeNPs were somewhat lower than the sums of inhibitions obtained with single ResNPs and SeNPs treatments, at least for low applied concentrations. This indicated that with respect to cytotoxicity, some antagonism between ResNPs and SeNPs was achieved in the composite, regarding normal cells. The variation in the effects observed between different cell lines could be cell-specific, depending on the available enzyme set in each line. Concerning the different activity of SeNPs obtained in A549 and MRC-5 cells, previously published data also confirmed higher toxicity of selenium-based nanoparticles to small lung adenocarcinoma [41]. Different selenium nanoparticles have also exhibited inhibitory effects against HepG2 cells [42,43]. On the other hand, in line with observed cytotoxicity of ResNPs against A549 and HepG2 cancer cells is previously published paper showing that resveratrol influenced cancer cells metabolism and induced apoptosis [44]. Furthermore, resveratrol was shown to induce apoptosis in A549 cells [45] and improve the anticancer activity of paclitaxel on HepG2 cells, at nontoxic concentrations [46].

To explain observed cytotoxicity, one could evoke the fact that nanoparticles applied in high concentrations induce physical cell damage, and consequently, inflammation with increased levels of reactive species, leading to oxidative damage of biomolecules [47,48]. In other words, if oxidative damage is accepted as a possible explanation of observed cytotoxicity, that would mean that the levels of produced reactive species saturated determined the antioxidative potential of the composite and its constituents; although some produced reactive species were neutralized due to antioxidative properties, some excess amount produced damage and consequently led to cytotoxicity. Indeed, some of the literature data suggested that at least part of the anticancer effect of SeNPs is based on their ability to induce high levels of oxidative damage, i.e., oxidative stress, which could be higher in cancer cells due to their low pH and already poor redox balance [49]. Moreover, not only the anticancer effect but also the antimicrobial activity of nanostructured biomaterials could be attributed to membrane disruption and oxidative damage generation in microbial cells, being proposed by the same authors [47,48].

Another important aspect of biocompatibility is the effect on the DNA molecule. Every new substance intended for human use should be evaluated in terms of genotoxicity because any damage to the genetic material in germinative or somatic cells can have significant consequences [50]. Trans-resveratrol alone, or in the mixed stilbene extracts, was not considered to be genotoxic, based on the previous studies [50] and our results, at least concerning normal MRC-5 cells are in accordance with previous data regarding this component. However, the nanostructure of resveratrol-based compounds, i.e., ResNPs, induced some genotoxicity in both cancer cell lines. Further on, there was no in vivo genotoxic effect in the lymphocytes of mice treated with selenium nanoparticles [51]. Still, we detected a low level of DNA damage induced by our SeNPs. Genotoxicity of SeNPs, observed in some samples, could probably be attributed to oxidative damage produced indirectly by physical cell structure disruption, as previously explained [47,48]. The observed lack of genotoxicity in MRC-5 treated with ResSeNPs was possibly due to the antioxidative effect of ResNPs, alleviating the oxidative damage produced by SeNPs.

Finally, since several different activities have been monitored, both beneficial (antioxidative, antibacterial/antibiofilm, and cytotoxicity against cancer cells) and harmful (cytotoxicity against normal cells and genotoxicity), it is of special importance to compare the ranges of active concentrations to validly estimate the ResSeNPs composite’s applicability. Comparison of concentration ranges of ResSeNPs composite applied in all performed tests pointed out that all beneficial biological activities, namely antioxidative, antibacterial, and antibiofilm, were determined in the non-cytotoxic ranges against all tested cell lines (≤1.5%). Moreover, all antioxidative, antibacterial, and antibiofilm composite concentrations were non-genotoxic in MRC-5 and A549 cells. However, the genotoxicity of its concentrations ≥ 0.78% requires caution.

## 5. Conclusions

The acquired results revealed the composite structure of ResSeNPs and highlighted its considerable potential as an antioxidative and antibacterial/antibiofilm material. FTIR analysis pointed out the involvement of hydrogen bonds in the composite structure, i.e., that the interaction between ResNPs and SeNPs takes place through the association of resveratrol with BSA, the stabilizing component of SeNPs. Further on, excellent antioxidative properties achieved by radical scavenging, lipid peroxidation-inhibitory, and ferric-reducing mechanisms, even at low concentrations, were determined. ResNPs and SeNPs components were responsible for the different mechanisms of antioxidative effects, with ResNPs being the main bearer of inhibition of lipid peroxidation, and SeNPs, together with the components presented within, having better reducing/scavenging abilities. SeNPs suspension was the main constituent responsible for high and moderate antibacterial effects against tested gram-positive bacteria (*Staphylococcus* species and *Enterococcus faecalis*, respectively). The antibiofilm effect against *S. aureus*, *S. epidermidis*, and *Pseudomonas aeruginosa* was pronounced at low concentrations, with higher activity of composite compared to single components. It is important to note that the promising overall activity of SeNPs and subsequently ResSeNPs can be attributed to all constituents found in the SeNPs suspension, which are, in addition to selenium nanoparticles, ascorbic acid and BSA. Also, it is of special importance that these active antioxidative and antibacterial/antibiofilm concentrations were non-cytotoxic and non-genotoxic in normal cells. Taking into account encouraging results on composite biological activities, our future research will be focused on further composite improvement with respect to physicochemical characteristics and improvement of biological effects to adapt it for more specific biomedical purposes.

## Figures and Tables

**Figure 1 nanomaterials-14-00368-f001:**
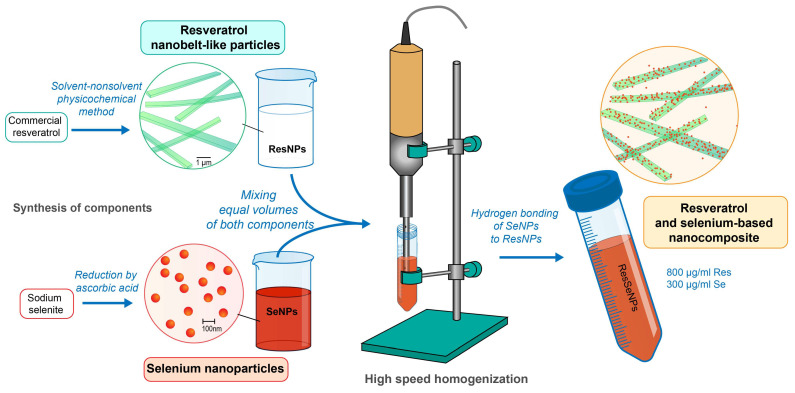
Schematic presentation of synthesis of ResSeNPs.

**Figure 2 nanomaterials-14-00368-f002:**
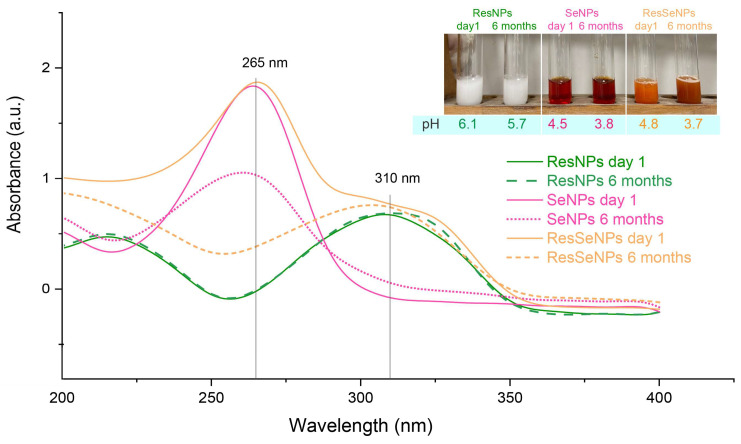
UV-Vis spectra of ResNPs, SeNPs, and ResSeNPs immediately after synthesis and following the 6 months of storage at 5 °C. Insert represents macroscopic observation of those suspensions, and their measured pH values.

**Figure 3 nanomaterials-14-00368-f003:**
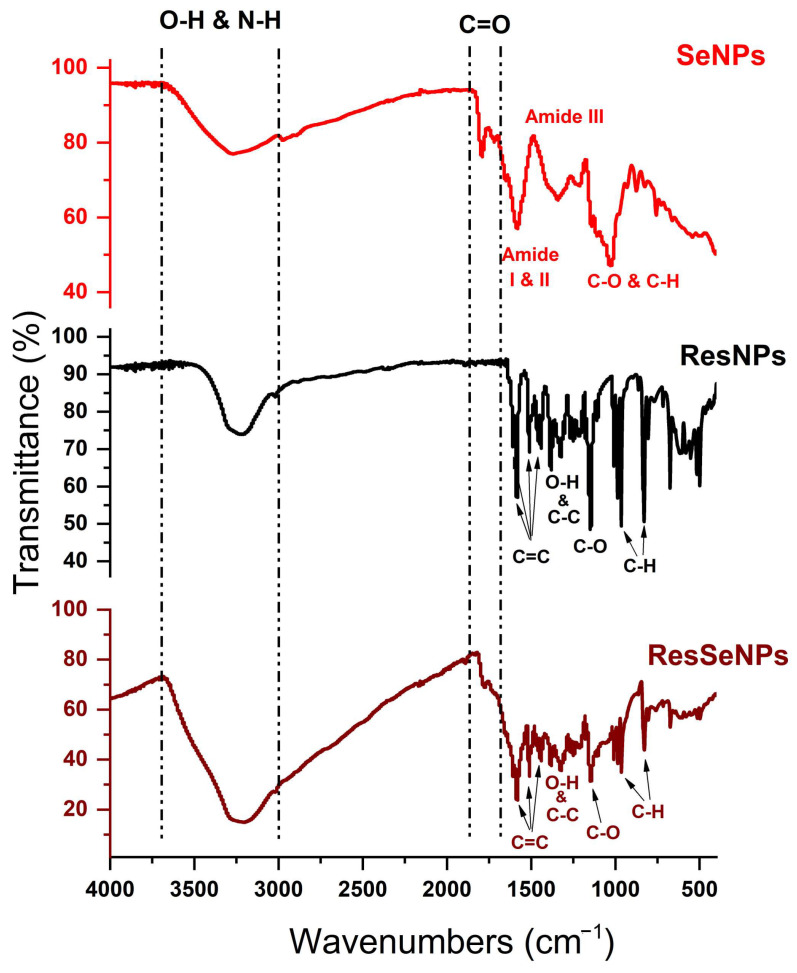
FTIR spectra of SeNPs, ResNPs, and ResSeNPs.

**Figure 4 nanomaterials-14-00368-f004:**
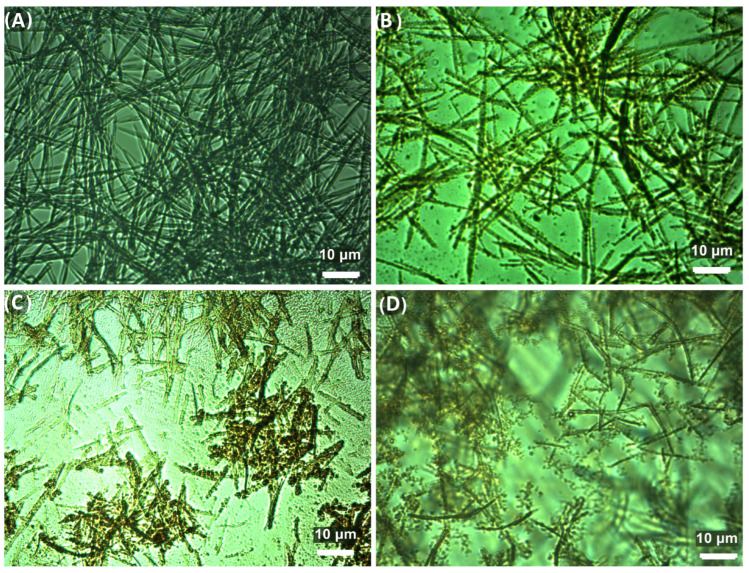
Representative light microscopy images, obtained under 1000× magnification, of undiluted (**A**) ResNP component; (**B**) ResSeNPs after synthesis; (**C**,**D**) ResSeNPs after exposure to 37 °C.

**Figure 5 nanomaterials-14-00368-f005:**
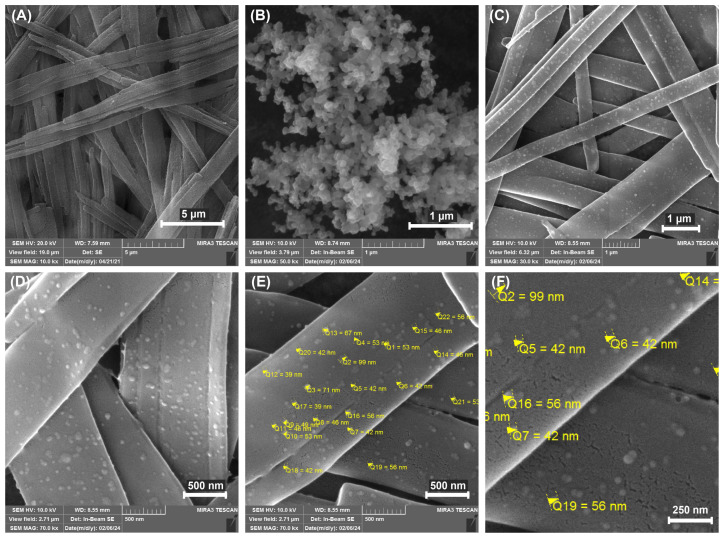
Representative FESEM images of (**A**) ResNPs, magnification 10,000x; (**B**) SeNPs from the original SeNPs suspension, magnification 50,000x; (**C**) ResSeNPs, under magnifications 10,000x, 30,000x (**D**) and 70,000x (**E**,**F**) a closeup of particle diameter measurements, respectively.

**Figure 6 nanomaterials-14-00368-f006:**
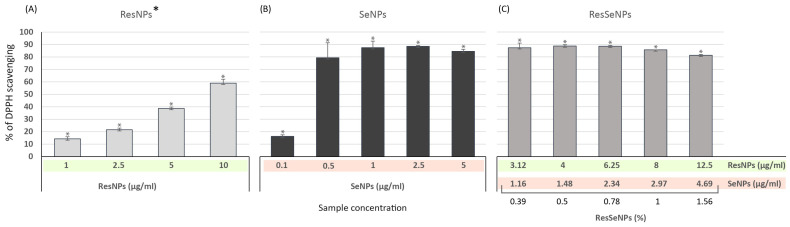
DPPH assay of (**A**) ResNPs, (**B**) SeNPs and (**C**) ResSeNPs. Values of the DPPH scavenging (%) by the test samples were calculated compared to the absorbance of the control DPPH solution that was used to set the 0% value, and the statistical significance was * *p* < 0.05. Tested concentrations of the ResSeNPs composite were as follows: 0.39, 0.5, 0.78, 1, and 1.5%. Labels on the *X*-axis are active concentrations of each ResNPs and SeNPs in the composite material. * Results of ResNPs scavenging activity are taken from our previously published research [11] T.

**Figure 7 nanomaterials-14-00368-f007:**
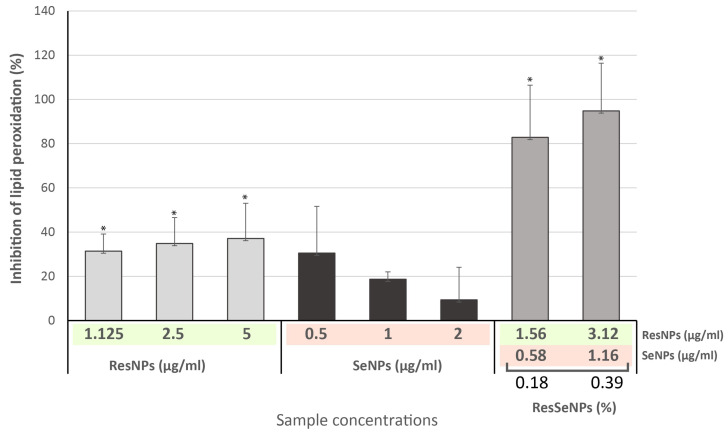
TBA assay of ResSeNPs. Results are expressed as the level of inhibition of lipid peroxidation (%) compared to the level of lipid peroxidation induced in the control sample (control sample values being 0%). The statistical significance threshold was * *p* < 0.05. Tested concentrations of the ResSeNPs composite were 0.18 and 0.39%. Labels on the *X*-axis are active concentrations of each ResNPs and SeNPs in the composite material.

**Figure 8 nanomaterials-14-00368-f008:**
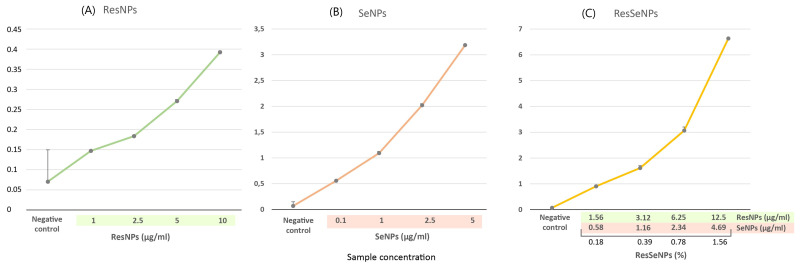
FRAP assay of ResSeNPs and components: (**A**) ResNPs, (**B**) SeNPs, and (**C**) ResSeNPs. Results are expressed as the absorbance values at λ = 700 nm, or the increase in A700 indicating the level of reduction of Fe^3+^ to Fe^2+^. Statistical significance was set to *p* < 0.05. Tested concentrations of the ResSeNPs composite were as follows: 0.18, 0.39, 0.78 and 1.5%. Labels on the *X*-axis are active concentrations of each ResNPs and SeNPs in the composite material.

**Figure 9 nanomaterials-14-00368-f009:**
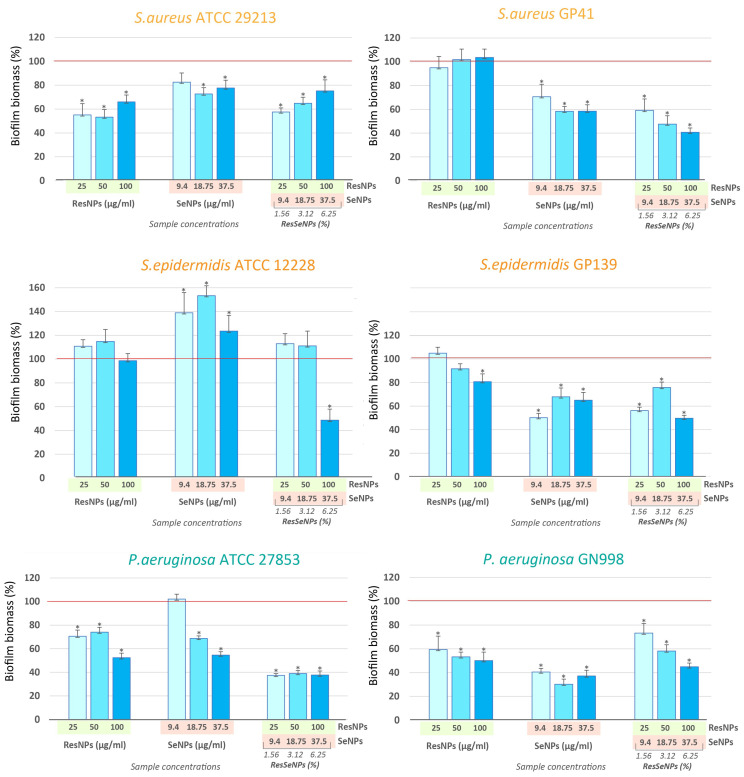
Antibiofilm activity of ResSeNPs and components. Results are expressed as the amount of biofilm mass (%) compared to the control samples used to set the 100% values. Tested concentrations of composite ResSeNPs were 1.5, 3.12, and 6.25% (*v*/*v*). Labels on the X-axes are active concentrations of single components (μg/mL). ResNPs and SeNPs were those equivalent to the composition of the tested concentrations of ResSeNPs. The chart showing the concentrations of components is at the bottom of Figure 8. The statistical significance threshold was * *p* < 0.05.

**Figure 10 nanomaterials-14-00368-f010:**
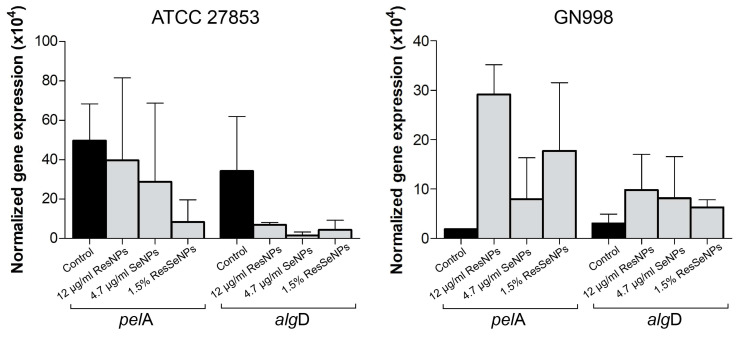
RT-PCR analysis of *pel*A and *alg*D expressions in *Pseudomonas* strains treated with 12 µg/mL of ResNPs, 4.7 µg/mL of SeNPs, or 1.5% of ResSeNPs. Results are expressed as normalized gene expression, and C represents gene expression in non-treated bacterial cells. The statistical significance threshold was *p* < 0.05.

**Figure 11 nanomaterials-14-00368-f011:**
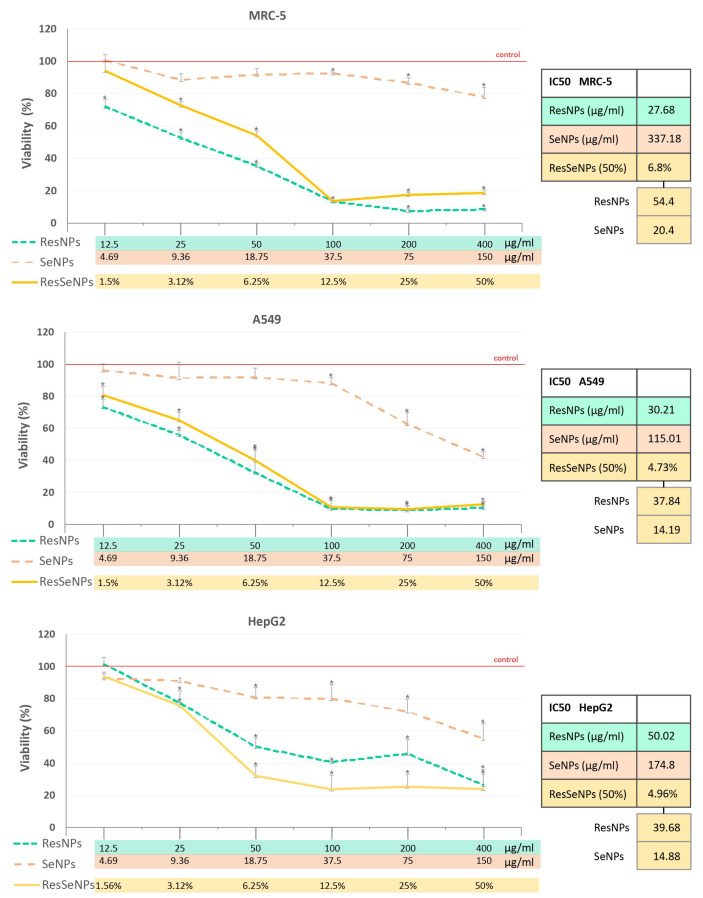
Cytotoxicity of ResSeNPs and components. Results are expressed as the cell viability (%) compared to the control cells whose absorbance values were used to set the 100% survival. Statistical significance was considered to be * *p* < 0.05. IC50 values are presented in the inserts. Values of ResNPs and SeNPs were equivalent to those contained in the tested ResSeNPs concentrations.

**Figure 12 nanomaterials-14-00368-f012:**
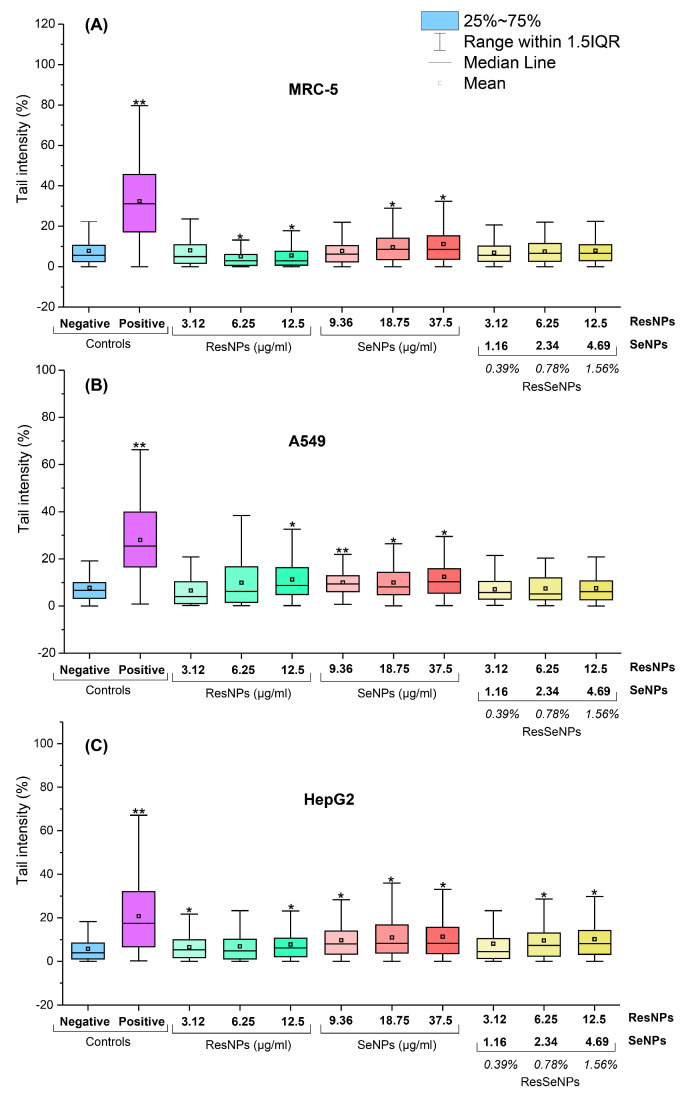
The genotoxic potential of ResSeNPs and components on different cell lines: (**A**) MRC-5, (**B**) A549, and (**C**) HepG2. Results are expressed as Tail intensity (%). Positive control cells were treated with 0.1% hydrogen peroxide. The boxes represent 25−75% of the obtained values, and the solid lines through the boxes are median values and significance * *p* < 0.05 and ** *p* < 0.01. The ResNPs and SeNPs concentrations equivalent to tested ResSeNPs are in the inserted chart.

**Table 1 nanomaterials-14-00368-t001:** List of bacterial strains used in this research.

Strain	Basic Traits	Origin
*Staphylococcus aureus* ATCC 29213	gram-positive cocci	ATCC collection
*Staphylococcus aureus* GP41	gram-positive cocci	Wound infection
*Staphylococcus epidermidis* ATCC 12228	gram-positive cocci	ATCC collection
*Staphylococcus epidermidis* GP 139	gram-positive cocci	Hemoculture
*Enterococcus faecalis* ATCC 29212	gram-positive cocci	ATCC collection
*Escherichia coli* ATCC 35218	gram-negative bacilli	ATCC collection
*Escherichia coli* GL289	gram-negative bacilli	Wound infection
*Pseudomonas aeruginosa* ATCC 27853	gram-negative bacilli	ATCC collection
*Pseudomonas aeruginosa* GN998	gram-negative bacilli	Wound infection

**Table 2 nanomaterials-14-00368-t002:** Primers used for RT-PCR analysis.

Genes	Primer	Sequence (5′-3′)	References
16S rRNA	Forward	GCAACGCGAAGAACCTTA	[19]
	Reverse	AACCCAACATCTCACGACAC	
*algD*	Forward	CGCCGAGATGATCAAGTACA	[20]
	Reverse	AGGTTGAGCTTGTGGTCCTG	
*pelA*	Forward	AGCAAGAAAGGAATCGCCG	[21]
	Reverse	GACCGACAGATAGGCGAAGG	

**Table 3 nanomaterials-14-00368-t003:** Free SeNPs in ResSeNPs suspension.

Sample	% Compared to Initial SeNPs Concentration ± Variation
Unbonded SeNPs from ResSeNPs after synthesis	34.62 ± 5.2
Unbonded SeNPs from ResSeNPs after 12h of incubation	40.75 ± 7.43
Unbonded SeNPs from ResSeNPs after 24h of incubation	32.65 ± 7.7

**Table 4 nanomaterials-14-00368-t004:** Antimicrobial activity of tested substances on gram-positive and gram-negative bacteria.

Bacterial Strain	ResNPs * (µg/mL)		SeNPs (µg/mL)		ResSeNPs ** (µg/mL of Active Constituents)				ResSeNPs ** (%)	
	MIC	MBC	MIC	MBC	MIC		MBC		MIC	MBC
Concentrations of Components in the ResSeNPs:					ResNPs	SeNPs	ResNPs	SeNPs		
*Staphylococcus aureus* ATCC 29213	800	800	4.69	9.37	12.5	4.69	25	9.36	1.56	3.12
*Staphylococcus aureus* GP41	800	>800	9.37	18.74	6.25	2.34	12.5	4.69	0.75	1.56
*Staphylococcus epidermidis* ATCC 12228	800	>800	18.74	300	6.25	2.34	400	150	0.75	50
*Staphylococcus epidermidis* GP 139	800	>800	4.69	300	6.25	2.34	400	150	0.75	50
*Enterococcus faecalis* ATCC 29212	>800	>800	75	300	200	75	400	150	25	50
*Escherichia coli* ATCC 35218	>800	>800	400	>400	400	150	>400	>150	50	>50
*Escherichia coli* GL289	>800	>800	400	>400	>400	>150	>400	>150	>50	>50
*Pseudomonas aeruginosa* ATCC 27853	>800	>800	400	400	>400	>150	>400	>150	>50	>50
*Pseudomonas aeruginosa* GN998	800	>800	400	400	>400	>150	>400	>150	>50	>50

* Results from our previously published manuscript [9]. ** The concentrations of ResSeNPs were expressed in two manners: as a μg/mL of each active constituent, and as *v/v* % of the whole composite in the medium.

## Data Availability

All data generated during this research will be available on request.

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
