# Peer review of "Resveratrol/Selenium Nanocomposite with Antioxidative and Antibacterial Properties"

_nanomaterials, 2024, doi:10.3390/nano14040368_

Round 1

Reviewer 1 Report

Comments and Suggestions for Authors

The authors highlight their studies concerning the Resveratrol nanoparticles, selenium nanoparticles and especially the combined Resveratrol/selenium nanocomposites. The detailed description of the obtention of each nanoparticle and the final nanocomposite is very well presented and as also all techniques in identifying/characterizing the compounds.

A lot of work has then been made and presented concerning the properties of the NPs and nanocomposite.

The data are very convincing, very well presented, and the conclusions wherever possible, are sound and  potentially important for the topic.

Author Response

Dear reviewer 1,

Thank you for the reviewing our manuscript. Please find our response in the attachment.

Best regards

Reviewer 2 Report

Comments and Suggestions for Authors

It seems that the Manuscript is well written. The main focus of the idea is also scientifically reasonable and very interesting. I suggest just increasing some light microscopy scale bars and numbers above. The MS can be accepted in its present form.

Author Response

Dear reviewer 2,

Thank you for the reviewing our manuscript. Please find our response in the attachment.

Best regards

Reviewer 3 Report

Comments and Suggestions for Authors

Dear Authors, 

I enclose my opinion on the manuscript submitted to the Nanomaterials journal. The research work is focused on the synthesis of resveratrol/selenium nanocomposite and the evaluation of antioxidative, antibacterial and cytotoxic properties. The manuscript is quite well written and structured. The aim of the authors is clear, but some results followed by discussion are neither convincing nor sufficiently discussed and require more work to be done. From an overall view of work, results and novelty I decided to reject this manuscript for publication in Nanomaterials due to the high standard of the journal. 

Further comments/recommendations and questions are below: 

  1. 1. The synthesis of ResNPs and SeNPs has been previously reported by the authors. However, readers should be informed in a short paragraph about the size, Z-potential, pH, temperature or storage stability etc. in the submitted manuscript. Similar information should be provided by the authors for ResSeNPs composite. 

  1. 2. The expected SeNPs release from the ResNPs surface should be experimentally confirmed or excluded. 

  1. 3. How do authors calculate the concentration of ResSeNPs composite in percentage? 

  1. 4. A few questions about SeNPs synthesis: Authors used ascorbic acid and BSA for SeNPs formation. On page 18 (494) authors pointed out that the presence of ascorbic acid in the suspension could contribute to the antioxidant activity of SeNPs. Did the author purify SeNPs nanoparticles after synthesis by centrifugation or dialysis? Moreover, did authors take account the antioxidative and antibacterial properties of albumin itself? 

  1. 5. Page 9: I recommend adding the SEM image of  SeNPs. 

  1. 6. Fig. 3, 4: The quality of all images is very poor and the coating is not clearly visible. I would recommend TEM microscopy for imaging all nanoparticles and composite.  

  1. 7. Fig. 7: What was used as the negative controlAuthors write about positive control in the methodology. 

  1. 8. The discussion should be more comprehensive. Authors should provide better interpretation of results in relation to the possible mechanismus of NPs and composite action, the size, geometry and stabilizing agent of NPs and composite, difference between free and attached SeNPs action etc. The correlation among assays and expected/inverse or different effect of NPs and composite should also be better discussed.

Author Response

Dear reviewer 3,

Thank you for the reviewing our manuscript. Please find our response in the attachment.

Best regards

Reviewer 4 Report

Comments and Suggestions for Authors

The current manuscript aims to report resveratrol/selenium nanocomposite with antioxidative and antibacterial properties. Although the topic is interesting in its scientific field, there are some issues that require the authors’ attention to improve the quality of this particular manuscript before further consideration for publication in a high-quality journal “Nanomaterials”.

Specific comments:

1.         The authors should carefully clarify the differences in the academic contribution points between the current manuscript and their earlier report (please refer to DOI: 10.1177/08853282231183109).

2.         Figure 1 is a schematic diagram of the synthesis of ResSeNPs. As stated by the authors, ResSeNPs were obtained by combining the suspension of ResNPs (adjusted to 1600 μg/ml) and colloidal solution of SeNPs (adjusted to 600 μg/ml of selenium, the stock concentration being initially determined by inductively coupled plasma mass spectrometry analysis). But, the audiences are unaware of the scientific meaning of 50:50. Please clarify.

3.         Please provide the optical microscopic image of SeNPs (Figure 3).

4.         Why the DPPH clearance rate may decrease as the concentration of Se NPs and ResSeNPs increases (Figure 5)? Please justify.

5.         Please explain the underlying mechanism of a significant inhibitory effect of “the combination of ResNPs with SeNPs” on the lipid peroxidation.

6.         In the antibacterial experiments, the authors simply provided quantitative evidences, but did not show the qualitative imaging data to support the reported numerical values. Please improve.

7.         What are the differences between ResSeNPs (μg/ml of active constituents) and ResSeNPs (%) in Table 3? Please further specify the scientific meaning.

8.         As stated by the authors, its low bioavailability has been an obstacle for practical use, leading to the development of nanoformulations to improve therapeutic potential. More recently, investigators have reported the development of resveratrol-based nanoformulations to improve drug bioavailability (please refer to DOI: 10.1021/acsnano.2c05824). In order to balance scientific viewpoint, the authors are highly recommended to consider the inclusion of this supportive case study in the reference list to enrich the research background.

Author Response

Dear reviewer 4,

Thank you for the reviewing our manuscript. Please find our response in the attachment.

Best regards

Round 2

Reviewer 3 Report

Comments and Suggestions for Authors

The authors properly answered the questions and recommendations. However, the antibacterial contribution of at least the ascorbic acid in the NPs solution should be shown experimentally, otherwise it does not correspond to the antibacterial properties of the prepared nanoparticles from the material and geometry point of view (question 4). Moreover, NPs release is the basic test that should be also reported in the manuscript (question 2). The authors should complete these experiments.

Author Response

We are grateful for the constructive questions and suggestions that have improved our manuscript in many aspects. Our response is in the attachment, with Reviewer’s comments rewritten in boldface, and changes in the manuscript listed with page numbers and marked by color coding (highlighted in turquoise).

Thank you, and best regards

Reviewer 4 Report

Comments and Suggestions for Authors

In my opinion, the revised manuscript is suitable for publication in “Nanomaterials”.

Author Response

Thank you very much, and best regards.